# Exploring children and young people's experience of participating in citizen science— A qualitative evidence synthesis

**Marie T. Frazer** [1,2,3]*, **Amy Creaser**[3], **Michael J. Tatterton** [1,4], **Andy Daly-Smith**[1,4]*, **Jen Hall**[1,2,3]

1 Faculty of Health Studies, University of Bradford, Bradford, West Yorkshire, United Kingdom, 2 Bradford Centre for Qualitative Research, Wolfson Centre for Applied Health Research, Bradford Royal Infirmary, Bradford, West Yorkshire, United Kingdom, 3 Bradford Institute for Health Research, Bradford Teaching Hospitals NHS Foundation Trust, Bradford Royal Infirmary, Bradford, West Yorkshire, United Kingdom, 4 Centre for Applied Education Research, Wolfson Centre for Applied Health Research, Bradford Royal Infirmary, Bradford, West Yorkshire, United Kingdom

* m.frazer@bradford.ac.uk (MTF); a.daly-smith@bradford.ac.uk (ADS)

**Data Availability Statement:** All relevant data are within the paper and its Supporting Information files.

## Abstract

### Objective

Citizen science with young people is becoming increasingly popular, and understanding their experience is valuable as it can improve research through improved participant motivation/ retention, alongside greater insight. The participants can benefit through opportunities to improve self-efficacy, learning, communication, and relationships. However, studies that explore young people's experience of participating in citizen science have not been synthesised.

### Methods

This qualitative thematic synthesis aimed to combine the literature on young citizen scientists' experience of participating in citizen science research studies. Seven databases, Google Scholar and The Journal of Citizen Science Theory and Practice were searched from 2012 to January 2022 and updated in May 2023. The screening included identifying articles by scanning titles and abstracts and, finally, full texts and selecting the articles using inclusion and exclusion criteria. The study findings were synthesised using inductive thematic synthesis (Prospero registration CRD42022299973).

### Results

Out of 3856 identified articles, 33 studies focusing on the participant experience were included in the synthesis. These papers were coded inductively. The resulting analytical structures were discussed and finalised. The researchers identified three main themes representing aspects of participant experience: relationships, power and personal growth, and three interwoven connecting themes: communication, self-efficacy and decision-making. An illustration of this would be communication bridging relationships and personal growth. As the citizen scientists' communication skills developed through personal growth, their

**Funding:** This study is funded by Sport England and the University of Bradford (match funding) via a PhD studentship. Sport England is a non-departmental public body under the Department for Digital, Culture, Media and Sport. The views expressed in this publication are those of the author(s) and not necessarily those of Sport England nor the Department for Digital, Culture, Media and Sport. The funders had no role in study design, data collection andanalysis, decision to publish, or preparation of the manuscript. Authors, A.D-S and J.H. were supported by Sport England's Local Delivery Pilot—Bradford; weblink: https://www.sportengland.org/campaigns-and-our-work/local-delivery (accessed on 24th January 2024).

**Competing interests:** The authors have declared that no competing interests exist.

relationships changed both with the project, with the researchers and with wider stakeholders outside the project such as school staff.

## Conclusion

These findings provide a comprehensive understanding of participant experience and how this can be used to inform future citizen science projects to facilitate a positive participant experience.

## Introduction

Citizen science is an umbrella term that describes various ways in which the public participate in science [1] and has the potential to transform society by empowering individuals and communities and contributing to breakthroughs in research [2,3]. It is a bridge between science and education, as it can facilitate improved understanding of important societal issues [4–6], and it can increase public engagement and involvement in research, improving researchers' understanding by reaching people and places and at scales not possible without citizen scientists' input [4,5,7,8]. Many participatory methods are encompassed within citizen science so it can be used across disciplines [9], with clear principles set out by the European Citizen Science Association (ECSA) [10].

Citizen science is a flexible approach that can encompass a range of different methodologies and methods, which enables researchers, practitioners, and policymakers to use the approach in a way that best fits their objectives [1]. Additionally, involving the public in research can help understand place-based meaningful issues people face, providing specific context and giving citizen scientists the knowledge and understanding to address these problems in ways that are acceptable to them [11,12]. There has been a rapid trans-disciplinary rise in research and public engagement in citizen science over the past two decades, with the method being utilised to improve scientific knowledge in fields as diverse as astronomy, physical activity, and water quality [9,11,13]. Natural science still dominates the citizen science landscape [14]. However, citizen social science is an established area of citizen science that looks at understanding social phenomena within the humanities and social sciences and can focus more on the social processes involved [15,16]. Albert [16] identifies a key issue for citizen social science is addressing the relationship between researchers and participants [16].

Participant experience is an important element of citizen science. Within this study a participant is a member of the public taking part in research. Participant experience includes the entire spectrum of thoughts, feelings, and perceptions that a person has when taking part in a research study. It encompasses all aspects of their involvement, from their initial motivation to participate, through the consent process, data collection procedures, and any follow-up activities. Participant experience is important for three main reasons:

1. To understand the benefits participants, gain from taking part as well as challenges participants face so these can be addressed [17],

2. Understanding participant experience will allow for improved participants' recruitment, motivation and sustainable commitment [18] and

3. If participants' expectations are met, scientific outcomes can be improved [16].

Previous literature has identified key factors influencing adult participation in citizen science that have been addressed, suggesting practical strategies for adult participation, e.g. developing communication strategies, and communicating results so the participant can see how they have contributed [14]. Such strategies could be applied to youth participation, but youth-specific citizen science research needs to be synthesised to support this.

The experience of young citizen scientists is particularly pertinent as citizen science is on the rise [11,17], perhaps best illustrated by the multinational 'You Count' project [18]; this project aims to investigate social inclusion and the challenges children face in relation to it. With the rise of such projects, understanding the young citizen scientists' experience and how this can be improved will be important to provide a positive experience for both the young people and the researchers. Young participants must be protected to 'uphold the merit and integrity of research' [19]. This protection must be balanced with the United Nation's (UN's) child's right to express their views on matters that impact them [20] so that children both have a say and are protected. Young people and children are defined as 24 and under [21]. Different standards are applied for children and young people, a consequence is that ethical regulations, for example, are different for young citizen scientists [22]. These different standards, combined with young people responding to different approaches, e.g. more creative data collection methods [23,24], create a different experience of being a citizen scientist, which should be investigated in their own right, so young people's experiences are fully understood.

Phillip's [19] comprehensive study of learning outcomes within citizen science and resulting framework identifies key elements of participation, including scientific knowledge and skills, self-efficacy, behaviour change and stewardship development; however, this research is not youth-specific and focuses on learning rather than the overall experience. There is no such review of young people's experiences. Current evaluative literature on the experience of youth citizen science participants focuses on online participation. These include iNaturalist [12] and Zooniverse [12] studies, which both focus on which elements of the project participants engage with for how long and how frequently, taking a quantitative approach to understanding participation.

There are some studies where participant experience as a whole and factors influencing the experience have been addressed through qualitative research as part of wider reporting on the citizen science study, but they have not been synthesised. These papers will be reviewed within this study.

## Aim and research objectives

This qualitative evidence synthesis aims to synthesise young people's experience of being citizen scientists participating in citizen science research projects. This will improve our understanding of what it is like for young people to take part in such projects, to inform future project design and help facilitate a positive experience.

## Methods

This synthesis reviews participants' experiences of being citizen scientists within citizen science research projects in different contexts. A qualitative synthesis was undertaken as qualitative methods allow for the exploration of the depth of the complex phenomena of human experiences. The review is registered with the International Prospective Register for Systematic Reviews (PROSPERO; CRD42022299973). Several terms describing citizen science were included in the search, as detailed in the inclusion/exclusion criteria. Using the term 'citizen' in citizen science can be considered contentious as not all members of the public may be citizens of the country they live in [20]. The term 'citizen scientist' in this synthesis is taken to

mean a global citizen and to distinguish the young people involved in research from the public not contributing to research and the professional researchers. This qualitative evidence synthesis was carried out in two parts. An original synthesis was conducted to form part of the first author's PhD study; an update was then carried out in May 2023 to incorporate more recent research.

## Literature search

A preliminary search indicated that young people's reflections were included in primary studies and specific researcher reflection papers. The researchers felt it would be appropriate to include both in the review to understand from different perspectives the participant experience. The research team (MF, JH, ADS, MT) and librarian developed the search strategy (S2 Table).

Searches were originally carried out on 21st January 2022 using several databases (EBSCO, MEDLINE, PROQUEST, Sportdiscus, Google Scholar, Pub Med and Scopus) (Fig 1). The Journal of Citizen Science Theory and Practice was additionally hand searched. The databases and additional searches were chosen to ensure coverage across disciplines. The searches were modified for the databases as required, e.g. Boolean logic was applied. Searches were repeated on 5th May 2023 for articles published after 21st January 2022; Sportdiscus was not included in the update as institutional access to this database was no longer available. The papers after

**Fig 1. Adapted Prisma 2020 Flow diagram.**

each search were imported to Endnote X9, which was used to locate and delete duplicates. The titles, authors, abstract and location information were then extracted to Microsoft Excel for the researchers to screen for inclusion and exclusion criteria.

There is to our knowledge no synthesis of young people's participation in citizen science projects making this paper the most recent search and synthesis of young people's experiences.

## Inclusion and exclusion criteria

The following criteria were used:

Inclusion criteria:

- The study's main cohort must be children or young people with a mean age under 25 years [21].

- The participants must be engaged in citizen science—The majority of citizen science principles 8/10 [10] must be evidenced within the study. Studies without open-access results were included, as constraints may not always make this possible.

- The study method must include a qualitative review of the participants' citizen science experience, either by the citizen scientists themselves or researchers.

Exclusion criteria:

- The research was solely quantitative.

- The research was undertaken for profit or if the research participants were paid more than a nominal amount or a stipend as this did not meet the authors' definition of citizen science [22].

- Studies published before 2012, with the rise in citizen science, and the accompanying literature e.g. the principles of citizen science being developed in 2015 [10] the authors felt the experience of participants before 2012 may not have been as relevant to today's youth.

- Studies not written in English.

## Selecting the literature

For the original search titles and abstracts, then full texts were independently screened for eligibility by two researchers using the inclusion and exclusion criteria. Where the researcher's view of whether a study was eligible conflicted with the other researchers, these studies were reviewed between the researchers, and where no consensus could be reached, a third researcher was consulted to reach an agreement. For the update, titles and abstracts, then full texts were screened by one researcher where there was uncertainty, inclusion was discussed until an agreement was met.

## Thematic synthesis

Researchers followed Thomas and Harden's three-stage approach to thematic synthesis using an iterative non-linear approach: i) line-by-line coding, ii) the development of descriptive themes and iii) the generation of analytical themes [23]. The strength of this inductive approach facilitates the discovery and interpretation of concepts from different contexts and across studies [23].

For the original search, all papers were independently line-by-line coded by two researchers. All sections of the research studies were read, and relevant information related to

participant experience was inductively coded in Nvivo version 12. Reviewers independently developed descriptive themes from their line-by-line coding. The researchers met four times as part of a reflexive process. 1) to discuss preliminary codes and descriptive themes and how they differed. 2) The primary researcher proposed four potential analytical structures, related to the aim, which were discussed, added to, and combined, to form a draft analytical structure. 3) The primary researcher presented the agreed analysis structure with coding that did not fit clearly into any theme discussed. The analytical structure was adapted. 4) Potential quotes for each section were discussed to illustrate the agreed themes.

The updated search papers were line-by-line coded by two researchers. Coders included researchers who had not been involved in the original synthesis to incorporate interpretation not tied to the original findings, alongside a researcher who had, in order to also relate the new material to the original findings [24]. The research team involved in coding the original and updated synthesis discussed and commented on the coding. Preliminary themes were identified from the updated search. These themes were then compared against the original analytical structure. Slight additions and changes were made before the reviewed structure was approved.

### Quality assessment

An adapted version of the Critical Appraisal Skills Program (CASP) [25] was used to assess the quality of the investigation of participant experience within the identified studies (S4 Table) The aims related to participant experience and the methods used to investigate them are displayed in this table. Where an item on the CASP checklist could be adapted to assess one element of the research, the wording was slightly altered, e.g. 'Was the research design appropriate to address the aims of the research?' became 'Was the research design appropriate to address elements of, or the whole participant experience?'. One item of the CASP checklist was removed (item 5): 'Was the data collected in a way that addressed the research issue?' as the researchers felt this could not be adapted to one element, often not the focus, of the studies —participant experience.

## Results

### Search results

The search is illustrated using an adapted Prisma flow chart: Fig 1. and reported using the ENTREQ guidelines S1 Table [26].

Out of 3856 identified articles, 33 studies focusing on the participant experience were included in the synthesis. In the original search undertaken in Jan 2022, 3095 papers were retrieved, 1197 duplicate records were removed, resulting in 1986 papers. After title and abstract screening, 168 papers were accepted for full screening (see Fig 1). Twenty-three papers met the inclusion criteria and were included in the synthesis. In the May 2023 update 761 papers were identified 190 duplicates were removed resulting in 621 papers. After title and abstract screening, 194 papers were accepted for full screening. (See Fig 1). 10 papers met the inclusion criteria and were included in the synthesis. After completing the quality assessment and discussion between researchers no studies were rejected based on quality (S4 Table).

### Study context and demographics

A description of the demographics of the young citizen scientists and research studies are available in S3 Table. 16 studies were conducted in the USA, with five in the UK, four in Canada and two in Australia. One study was conducted in each of the following countries: the

**Table 1. Codes shown by types of citizen science and main themes/ aspects.**

| Type of citizen science | No of papers | No of codes to | | | | | | Aspect | |
|---|---|---|---|---|---|---|---|---|---|
| | | Themes | | | | | | | |
| | | Power | Relationships | Personal growth | Communication | Decision making | Self-efficacy | Factors influencing | Wider impacts |
| Action Research | 16 | 34 | 174 | 270 | 26 | 8 | 24 | 75 | 47 |
| Co-Production | 3 | 5 | 20 | 20 | 12 | 2 | 0 | 39 | 8 |
| Community | 3 | 2 | 24 | 50 | 5 | 1 | 0 | 13 | 6 |
| Community Based Action Research | 3 | 8 | 10 | 49 | 3 | 1 | 2 | 22 | 8 |
| Other | 4 | 17 | 44 | 68 | 17 | 1 | 11 | 23 | 10 |
| Participatory Research | 4 | 20 | 27 | 28 | 7 | 3 | 5 | 27 | 11 |
| Total | 33 | 86 | 299 | 485 | 70 | 16 | 42 | 199 | 90 |

Philippines, Hong Kong, New Zealand and the Caribbean. One study [27] involved multiple countries (Bangladesh, Bosnia and Herzegovina, Brazil, the Democratic Republic of Congo, Mongolia, Romania, and Sierra Leone) and one [28] was carried out across European countries (the Netherlands, Norway, Poland, Portugal, and the UK).

The study topics varied and were from different disciplines, e.g. conservation science education [29] and indigenous involvement in community research [30]. S3 Table categorises the disciplines the citizen science was conducted in; overwhelmingly, the studies came from social science or humanities. Young people's participation in the research was labelled in several ways: e.g. Youth Participatory Action Research (YPAR) [31], participatory research [32], community-based participatory research [33] and co-researching [34]. Table 1 summarises the types of participation and the number of papers (files) of each type. The most predominant type of citizen science was action research. The age and demographics of the children and young people involved also varied, with an age range of 10–25 years with most young people being teenagers- mid-twenties (S3 Table).

## Quality assessment

No studies were excluded based on quality. (S4 Table) However, the assessment did highlight some areas where the participant experience was not reported on adequately. Within some studies the participant experience was not identified, even briefly, as an aim or objective of the study, rather it was an added discussion, raising concerns about the appropriateness of the research design to address participant experience in some studies. However, it has been reported that within qualitative synthesis, excluding inadequately reported studies will not greatly impact the findings [27].

## Thematic synthesis

From the original line-by-line coding, six themes were identified. Three themes were considered overarching; power, relationships and personal growth, with three themes considered interlinking; decision-making, communication and self-efficacy. Two further aspects connected to this core experience of the participants were identified: 'factors that influenced the experience' and 'wider impacts' of the experience beyond the research. These themes did not necessarily constitute participant experience, but were identified, through inductive analysis, as key aspects that impacted or were impacted by participant experience. The breakdown of codes to key themes and aspects can be seen in Table 1.

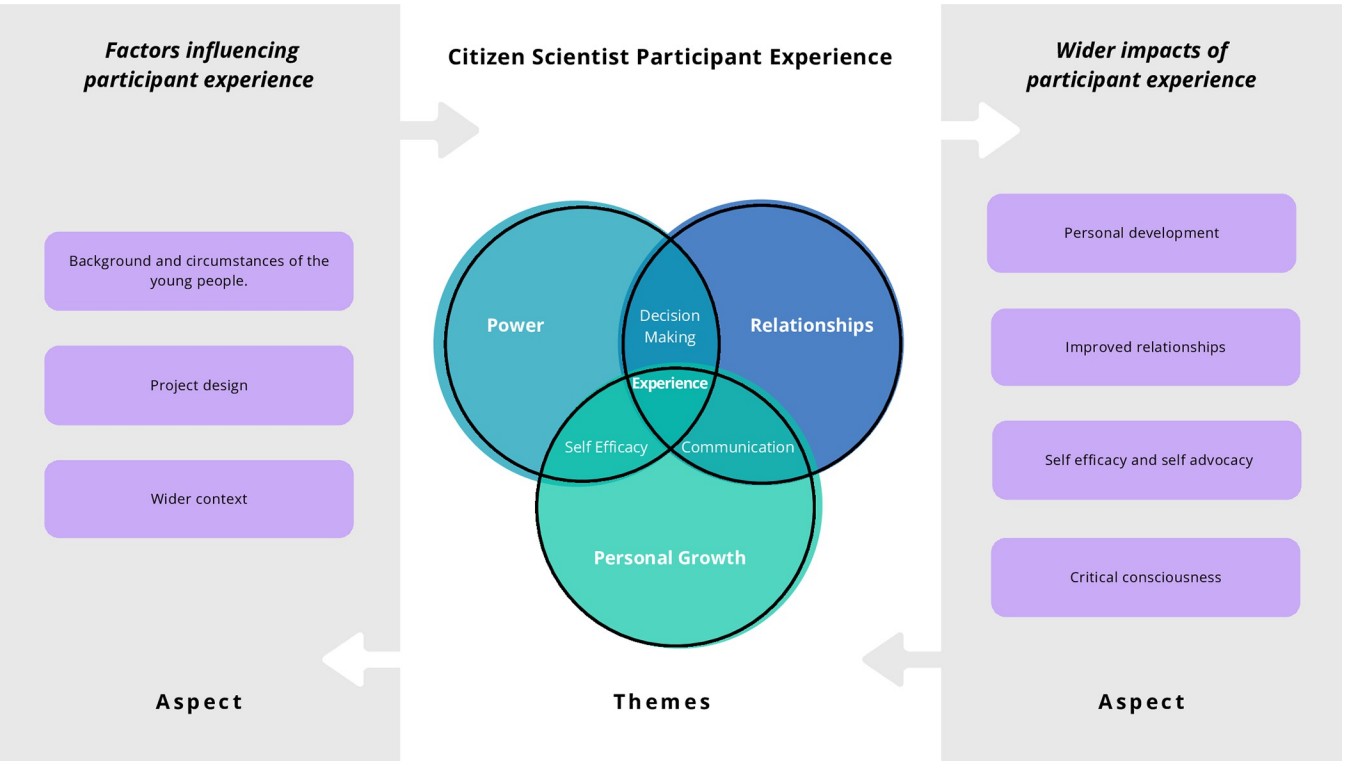

**Fig 2. The themes and aspects relating to participant experience.**

From the updated synthesis, despite anticipating that new themes would be found due to the additional papers, and new researchers' interpretations, the codes created a similar structure with further detail being added to some themes and slight changes being made to the title/definition of others. How the themes and aspects interlink is illustrated in Fig 2.

The findings below describe each of the three main themes, three interconnecting themes and two aspects related to participant experience in turn. While themes are described separately here for clarity, they can also be viewed as intertwined threads that together create an understanding of participant experience.

Whilst focusing on one theme, it is important to situate it in the context of the wider findings, including 'factors influencing' and 'wider impacts', to understand the complex tapestry fully. The 'factors influencing participant experience' and the 'wider impacts' are discussed below. As an overview, we categorised the factors influencing participant experience as a) the background and circumstance of young people, b) project design and c) wider context; see Fig 2. The background and circumstance of young people encapsulates their knowledge and skills alongside personal characteristics such as gender and ethnicity. Project design factors included space for reflection, accessibility, and time management. Wider context encompasses factors such as COVID-19, political tensions, community support, access to technology, etc., within the external environment in which the research projects operate.

The wider impacts of participant experience refer to aspects of the experience that could stay with the citizen scientist after the research project and affect the broader community. The key elements of wider impacts, highlighted in Fig 2. include personal development, improved relationships, self-efficacy and self-advocacy, and critical consciousness.

## Personal growth

This study defines personal growth as developing an individual's skills, knowledge, confidence, and emotional development. Citizen scientists experienced personal growth through participating in the research [28–30,32,33,35–45]. Multiple studies found that young people experienced increased knowledge relating to the research topic [29,32–35,37,39,42,44–48]. This subject knowledge varied from 'the state of black girls' education' [41] to relationships between sand crabs [29] and understanding of health inequalities [35,48]. Within some studies [29,32–35,37,39,42,45,47,48], topic learning led to some participants becoming 'capable of critical and self-reflective practices' [47] as they reflected on what they had learnt. The participants' understanding of research changed from being something you 'looked up on a computer' to a 'bigger process' that included 'ideas and behaviours of real people' [31].

Projects provided citizen scientists with the opportunity to develop research skills [28,29,31–33,35,37,39,41,49], including; writing skills [47], communication skills [28,50,51], presentation skills [38], problem-solving [31,32,35,37,39,41,48], leadership [33,44]; and method-specific skills, e.g. ethics, question styles, and interview protocols [42,46,51].

Some participants experienced personal changes such as building personal resilience and empathy [40,42,52,53].

Bennett [42] reported participants:

'noted that reading numerous message threads across the Childline Message Boards, during the initial familiarization exercise helped develop their understanding and empathy'. p3151 [42]

A facet of their experience was social learning, this included participants learning about themselves through reflecting on their observations of other individuals and wider society. [29,31,41,47,48,52,54].

Citizen scientists reflected on how they learnt of others' immigration stories and how this led them to reflect on their own.

'We learned about (his) story and how his family came here from Italy" and that "(his) family save[d] his grandfather's recipes." Throughout the project, EB students also became more curious about their own migration stories, for example, "since I come from an immigrant family it makes me want to know more about my family that I didn't know before.' p433 [38]

Social learning differed within studies. In Abraczinka's [53] paper, participants in the YPAR and physical activity programme had a positive experience stating:

"It helped me learn things that I can do to make the community better and more fit." P237 [53]

In a juxtaposition, some participants started to understand the system is weighted against them; here the social learning begins to blur with self-efficacy and empowerment. Participants taking part in Abraczinkas' [53] YPAR only study did not believe their opinions mattered or that 'change will actually occur'.

The literature discussed suggests that as citizen scientists acquire skills, knowledge, confidence, and emotional understanding, their capacity to communicate and interact within relationships undergoes a transformation. This, in turn, influences the power they perceive and wield, as evidenced by their decision-making.

## Relationships

In this study, relationships are defined as the connections and interactions between individuals. The positive meanings associated with these human connections contribute to a favourable participant experience [42,52]. The meeting of new people and the expansion of social circles, meeting people with different experiences, backgrounds, and skills, enhanced the participant

experience [21,33–35,37,38,40,41,43,47–52,54,55]. These new relationships caused participants to reflect on their place in the world [31,32];

'youths' reflections in regard to changing their own stigma around homelessness and shifts in identity towards change agents could serve as potential motivators for exiting the streets.' p386 [32]

In others, it led to feelings of connection with their community or people within the research [32,34,38]

"Being on a team and knowing that I'm a part of something and other people. It's not just all me." p220 [43]

Participant experience was most affected by the citizen scientists' relationships with three key groups: 1) with the research facilitators, 2) with (citizen scientist) peers, and 3) with wider stakeholders, as outlined below.

Developing relationships and connections with the facilitators was a key part of the participant experience [30,32,34,35,41,51]. Feelings of 'respect', 'appreciation' and 'feeling valued' [32,42,46] were characteristics of positive relationships between facilitators and participants. Ford [30] emphasised the importance of establishing trust between the facilitator and citizen scientist. A citizen scientist describes below how this relationship can influence participant experience, including personal growth:

'At the beginning, I was nervous because I did not know if I am going to do well with my interviews. As we started working together with the adult researchers, everything was relaxed. They were easy-going and understanding with our limitations. . . I feel we became friends with them, and I am not shy anymore. I am so happy as this was an amazing chance to be a researcher and to learn from other researchers.' p6 [34]

In contrast, within the same study another citizen scientist felt 'pressured' by the relationship with the facilitators [34]

'I was reading and reading to try to find something to give feedback. It was stressful. I found something to correct, but I was hesitant to say it or not to say it. That was a lot of pressure. I think my problem was that I wanted to give smart opinions, and I wanted to say something that the researchers will be impressed with.' p6 [34]

The citizen scientist-facilitator relationship was 'bidirectional', [32,35]. Not only did the participants acquire skills and knowledge (as described in the personal growth section above) but the researchers and facilitators also learnt from the experience, as illustrated by the adult coordinator below:

'Most definitely putting myself in the student role at times because I learned so much from them. This allowed me to be a student listening to them about what's out there, what's important, what they got to their peers. Introducing me to what information they collected. That right there is huge'. (Adult coordinator, Chicago) p719 [35]

Relationships between the citizen scientists and facilitators (researchers) were interwoven with power dynamics [32,34,54]. The following quote illustrates how the young citizen scientists (co-researchers) began as learners, and then, as the relationships developed, they took the role of experts.

'The co-researcher/researcher relationship began during training sessions where university researchers took the lead and co-researchers were positioned as learners. As the project progressed, co-researchers increasingly assumed the role of community expert, with university researchers becoming learners.' p6 [54]

Peer-to-peer relations were an important part of the participant experience. A citizen scientist in Warren [31], conveys the ideas of social support and a sense of belonging created by these relationships formed in the research project:

At the beginning I thought some people were kind of mean . . . now we're all one big family and I mean, we can ask each other advice. We depend on each other. We have a group chat that's called 'research sibs'. p697 [31]

This notion of developing belonging is supported by Scott [47]. As the research team spent more time together, the young citizen scientists began to develop social bonds through the process of carrying out the research. Scott [47] describes this as 'more than friendships' as, through the research, the citizen scientists recognised themselves in the stories they were hearing as part of a marginalised group. This gave them the opportunity to 'rethink belonging and challenge simplistic notions of diversity.' [47].

The young citizen scientists also described relationships with the people they conducted research on as forming part of their experiences [34,51,54,56], ranging from an interest in the research subjects' stories [34] to understanding that the citizen scientists and the research subjects have 'personal connections and shared community and/or cultural knowledge' [54]. The young citizen scientists in Goto [56] saw the importance of these connections in discussing HIV/Aids:

'We have professionals that are very judgmental. "You should not be having sex, you should go to church." It is better to train young persons.' (Youth researcher) p401 [56]

This close connection between the citizen scientists and research subjects in Braye [51] led to a blurring of the citizen scientist/research subject experience, creating frustrations between the citizen scientists and university research facilitators. For example, the citizen scientists didn't want to pry into the research subjects' lives around sensitive topics whereas the facilitator wanted this level of detail. The young citizen scientists also wanted to offer the research subjects advice which was counter to the advice given by the facilitator. The facilitators observed that the 'identity lines between researcher and researched were much less clear for our peer researchers' [51].

Developing or evolving existing connections with the community was also part of the citizen science experience [30,32,45,54]. Some participants witnessed a change as the community members started to view the young citizen scientists as researchers and relationships altered.

'. . .cause most teachers see me as like, a goofy kid in the class, but once they actually got to see me in this position, they were like, "Oh there's some potential in that student.' p277 [45]

The peer, research and community relations mentioned above often interweave and affect each other. Positive relationships with the facilitators and peer researchers often supported each other and led to positive relationships with the other stakeholders.

**Power.** This synthesis defines power as having the 'capacity and opportunity to fulfil or obstruct personal, relational, or collective needs' [57]. Empowerment is noted as part of the participant experience in most studies [28,30–33,35,37,40,44,48,49,51,54,56,58]. Citizen scientists felt they were empowered by becoming citizen scientists in a number of ways, e.g. by being part of the group decision-making and feeling listened to and understood [32,39]. In Martinez's [35] study, creating a curriculum that will authentically engage youth of colour, enabled youth to feel 'an increased sense of agency and ownership over the project'. Warren [31] study into urban education reform describes how the citizen scientists were able to use this agency developed through the citizen science project to 'conduct independent research as a tool to both recognise injustice and make recommendations to policymakers necessary to redress education inequity.'[31].

Some citizen scientists experienced a change in power relating to their community, they had more influence on community members, or as Bender [32] describes it the citizen scientists became 'change agents' for their communities. A participant in Lane's [49] study, through the knowledge and research skills acquired from taking part as a citizen scientist, was able to influence the choices of other community members.

'You know what the highlight of my day was? I was talking to this lady with her daughter. They had a bunch of pop in their cart and I talked and handed her the flyer and I saw her reading it . . . she said her daughter had just got diagnosed with diabetes . . . then when I saw her later she had put back a bunch of the pop!' p265 [49]

A participant in the LiMPETS (Long-term Monitoring Program and Experiential Training for Students) study explains:

'I never would have imagined that such a young generation could communicate with, like, the older generation who have accomplished so much, and just. . .kind of be on a somewhat level playing field. Like they'll respect us despite being kids. . .' p72 [29]

The quotes above show how the participant experience can include changing power dynamics within the community where the participants' voices are listened to.

Some citizen scientists reported negative experiences related to power [38,43,46,53]. Within Abraczinskas' [53] health equity study, the participants had different experiences depending on which project they had taken part in. In the YPAR-only study, the stakeholders with decision-making power were not invested in the project. Despite the research project and the learning and skills developed by the citizen scientists, they didn't have the power to effect change. One of the outputs of the citizen science work, a request for an outside basketball net to be straightened, was not put into practice [53]. This experience was also noted by research facilitators [38].

'We found that I–O students' desire to create social change in their schools was dampened when they encountered peer and teachers display disinterest' p10 [38]

Canosa [28] describes how the citizen scientists in their project opted out of the analysis phase of the research as they viewed this as the research facilitator's job. This shows how power in this relationship led to the citizen scientists taking a step back from the research rather than assuming further ownership. In other studies [35,40,54], participants took further ownership and control over the research:

'Throughout the process, power shifted gradually from the adult facilitators to the youth, who took charge in planning their actions, implementing them and facilitating the parents' meetings. The participation level raised from 'consulted and informed' to 'adult initiated and shared decisions with children' p420 [40]

**Self-efficacy.** Through our synthesis, we positioned self-efficacy as a link between personal growth and power. Self-efficacy is where participants, through the research, developed an understanding that they had the ability within themselves to affect change for themselves, as illustrated by this participant:

'even though we are students and minorities, we can make a change.' p280 [45].

Positive changes in participants' self-efficacy, linked to personal growth are detailed across several papers [32,33,37,38,44,49]. The success of the research can also increase a participant's self-efficacy. This experience of self-efficacy was inseparable from the citizen scientists' growing confidence and resilience or 'tenacity to achieve their goals.' [32]. Toraif [37] additionally highlights that taking part in a research project can impact efficacy negatively; as young participants further understand systemic racism and racialised dynamics, their sense of efficacy can be reduced.

**Communication.** We have defined communication as the way information, feelings or ideas are conveyed or shared between individuals or groups, this can be through different means, e.g. visually or verbally. Many studies [29,31,32,35,38,42,44,49,53,54] refer to communication among the citizen scientists and other stakeholders. Participants mentioned improved communication skills [29,32,35,49], which in several cases led to the confidence to communicate with people beyond their social circles, for example, the general public [35], school representatives [45] and the scientific community [29].

The citizen scientists are involved in various activities such as public speaking, presenting and engaging in research, contributing to their overall communication development. We see this conveying of ideas as a connector between personal growth (where participants' communication skills improved) and relationships (which led to a change in those relationships):

'. . .. pushed me to step out my comfort zone and socialise more and I was able to meet new people and see how different neighborhoods around my city were living.' (Youth participant) p718 [35]

Here the young citizen scientist, through communication, has made new relationships providing them with a different perspective and a better understanding of their context.

Within Genius [54], citizen scientists used different communication techniques to encourage conversations with research participants, for example:

'Co-researchers also built rapport with the interviewees by articulating personal connections and shared community and/or cultural knowledge ~ for example, recognition of shared geography, built environment, and/or people.' [no pagination] [54].

This two-way exchange where information is volunteered and then a prompt is given for a response, was identified as a way to gather information from indigenous people. This illustrates one way participants own experiences can enhance communication, to create a unique opportunity for the citizen scientists to connect with participants [54].

**Decision making.**   Several studies reflected on how decisions were made [28,32,35,37,39,44,51]. Within our analysis, decision-making is conceptualised as a connecting thread between power and relationships [32]. Canosa [28] argues the 'biggest barrier to children's participation is the need in every case for an adult to consent to their participation' [28]. Where parental consent has been sought, the power a child has to become a citizen scientist or not could be taken away. As a counterpoint, children do have the power to stop taking part in the research, against the researchers and/ or parental preference which does afford the young people a degree of autonomy. An example of removing autonomy is described in Budin's [46] work: Participation here in some circumstances was obligatory if the young people wanted to achieve their diploma which led to frustration.

Davis [41] describes how a participant left the research despite frequent approaches from the researchers to rejoin as she didn't think the project was beneficial to her. Personal growth, including improved communication skills and confidence, altered self-efficacy and power dynamics. This in turn led to citizen scientists being more involved in decision making processes as part of the research [49].

## Factors influencing the participant experience

Within Fig 2. a summary of the main factors broadly influencing participant experience are presented. Factor influencing participant experience refers to various elements that impact how individuals engage in different activities or studies.

The various themes and aspects that have been examined are tightly interconnected. A notable addition to the findings from our update is the COVID-19 pandemic as a significant 'wider context' factor shaping participants' experiences.

Budin [46] highlights how a shift to completing research online unintentionally excluded participants who lacked digital access. This unfortunate reality underscores the unequal power dynamics that emerged during the pandemic, magnifying disparities and impeding the personal growth of those unable to fully engage. In a contrasting perspective, Song [44] presents COVID-19 as an unexpected way for young people to act as catalysts for positivity in their community, fostering the development of leadership skills and, consequently, personal growth.

COVID-19 also impacted relationships, for example, virtual gatherings held as part of the research also served as a space to mark birthdays and other milestones as a group, which may not have occurred to the same extent had the young people not been in lockdown [36]. Similarly, Cooper [50] reflects on the citizen scientists using emotive terms in their work due to the public health restrictions, which meant they could not be together in person.

The remaining factors influencing participant experience are described below in relation to the main themes of power, relationships and personal growth.

## Factors influencing personal growth

The citizen scientists' backgrounds influenced their participant experience [33,35,42–44,47,48,55,59]. This was particularly evident in the work of Song and Hur [44], where citizen scientists not proficient in the Korean language encountered distinct challenges that diverged from those faced by their language-fluent counterparts. These challenges, in turn, left an imprint on their development throughout the course of the research study. Song and Hur [44] also describe how their participants went through a selection process where participants were graded on past leadership experience, with those scoring the highest selected for the project, which could have impacted the young people's ability to put their ideas into action successfully and demonstrate leadership skills throughout the project.

Training of the citizen scientists was reported to influence personal growth, particularly learning and skills development. In many studies [28,30,34,41,45,51–53], training is incorporated within project design (Fig 2). as a key aspect influencing experience. Training consisted of planned activities for the citizen scientists, often delivered in workshops [29,30,32,34,38,41,49,58]. Davis' [41] study includes a table of activities used and their purpose; for example, the workshop entitled 'Going Back in Time and Looking at the Future' included identifying educational issues that were important to the citizen scientists. A further example is taken from Lane [49], where they use a role-play activity to aid confidence development.

Planning for, including and encouraging reflection was evident across the studies [32,38–41,45,47,51–54,56], which was reported to lead to improved personal growth. In Fowler's [39] study, reflection was planned to focus on the citizen scientists' skills development and learning needs. Genuis [54] and Fowler [39] facilitated reflection through photovoice, where the citizen scientists were paired with university researchers to reflect on their research and how to improve it.

Some studies highlight a lack of time as a barrier to citizen scientists' research development and personal growth [31,42,46,50–53]. Time pressures were the result of already busy lives overlaid with additional research time [28,32,41,47]

## Factors influencing relationships

The research design was integral to the relationship facet of participant experience [32,35,51,52,60]. Shea [52] illustrates this:

'The lead-up to our project with the girls consisted of a number of meetings and conversations, group planning, and collaborative activities, which in turn contributed to relationship building.' p278 [52]

The design, including planned activities, created opportunities to foster relationships. Exercises such as problem-solving helped to 'build group norms around communication and decision-making' [35] and 'establish collective expectations' [32]. The citizen scientists also saw these activities as central to

'cultivating the types of relationships needed to (a) feel comfortable to engage in dialogue and reflection and (b) confront sensitive topics including racism, ageism, and other forms of oppression and/or discrimination.' p488 [37].

Relationships can take time to develop. Bender [38] recognises 'true engagement' took some time. Toraif [37] acknowledges:

'adult allies and youth members both underscored the need for ongoing dialogue and reflection beyond initial project preparation and orientation. Youth discussed the importance of conversation in building trust, rapport, and community with adult allies and among the youth members' p487 [37].

This discussion followed a seven-month data collection period, and although they do not link these elements to time specifically, having multiple opportunities within this timeframe to create and foster relationships could have developed trust and rapport.

Wagaman [60] and Bender [32] identified shared vulnerability as a key to relationship development, and referred to a 'safe and brave space' as being necessary to allow personal sharing and vulnerability. Training may be needed to encourage this safe space as one young citizen scientist observes about their facilitator:

'I feel like they mean well, but I don't think they. . . it doesn't seem like they've worked with youth of color before (Youth Member 8)'. P477 [37]

To build and sustain relationships using technology as part of the communication within the research was identified as useful [31,42,46,48], with Warren [31] highlighting 'GroupMe chats' as 'the most useful communication modes for youth to engage with one another' with regards to the research, and Akom [48] proposing that:

'using ICT, Location-based, wireless, digital media, and participatory forms of technology also allowed youth from different cohorts to digitally organise and communicate even when students had never physically met in person.' [48].

Seeing the value of the research, and the participants feeling that they meaningfully contributed, positively influenced their experience [33,40,42,44,46,47,55]. A citizen scientist in Bennet's [42] study highlights this:

'I enjoyed feeling that I had made a difference and helped improve the service. The Childline/NSPCC session was great. It felt that as we were talking to people who were really closely tied to the process of improving the boards that what we were doing was really helpful and directly contributing.' [42]

Here, experience was positively impacted through the relationships between the childline staff and the citizen scientists as they were able to see that they were directly contributing.

Marciano [43] highlights how the school children attend had an impact on the young peoples ability to carry out an analysis task, as schools varied in permitting access to data.

'when it was not possible for all members to contribute data from their respective schools equally, tensions were raised within the group that influenced whether and how the students were able to engage in data analysis.' P219 [43]

A further factor that needs consideration is the personal traits of the citizen science participants and how this impacts the relational aspect of experience; for example, MakerCastero [38] states: 'self-reported introverted students report feeling shy and awkward in engaging peers in the EB classroom.'

## Factors influencing power

External expectations of the citizen scientists led to changes in power, illustrated below [29,31,54]:

'nearly all the youth described ways that being accountable to these external audiences drove them to learn new content and skills, take on new roles and expertise, and realise how those skills gave them power to take action.' p72 [29]

Other studies [29,31,33,40,42,44,50,61] also suggest that a safe space and adequate support led to feelings of empowerment.

'Findings reveal that this project provided the space and adult support necessary to empower these 15 youth to identify worthwhile areas of inquiry and, as a result, use their voices to collaboratively determine outcomes that might somehow change the function of their schools to better serve marginalised youth (and families).' p694 [31]

Wagaman [61] discusses the actual physical space, what that can symbolise and how it can influence the participant experience.

'Upon reflection, there was power that the team gained by meeting in a space that was quiet, physically accessible, and positioned the team such that they could look out over their physical community as they discussed their goals for impacting it." p87 [61].

But this power only came with time, as the citizen scientists built a sense of ownership of the space [61].

How activities within the citizen science projects were designed and implemented affected the experience of power. Bender [32] notes that 'power-sharing in group activities was a critical strategy.' Facilitators structured activities with collective expectations in mind. Activities that linked the citizen scientists and people from outside the research gave opportunities for joint power development; for example, Abraczinkas [53] created a 'feedback loop between youth and school stakeholders' which resulted in the principal of the school asking the citizen scientists' opinion, which could lead to increased collective power.

## Wider impacts on the citizen scientists taking part in a research project

This section will discuss the wider impacts of participating in a research project on citizen scientists which encompass the significant effects that individuals' engagement in research can have both on the participants themselves and the broader community. Here personal development, critical consciousness, self-efficacy, self- advocacy, and improved relationships are suggested to go beyond the confines of the citizen science studies.

In terms of personal development, career and college readiness was identified in several studies [32,35,45,47,51] as a personal change that could have a wider impact on young citizen scientists. This varied from the citizen scientists being able to say they have taken part in an extracurricular activity to the 'professionalisation' of the students via their training in research methods, [45], which in turn led to the citizen scientists being seen differently by their peers and other stakeholders.

Some participants developed a critical consciousness, defined here as an ability to recognise injustices, summarised in the quote from Martinez [35]:

'By incorporating the diverse experiences of youth of color through a critical youth empowerment and critical race theory, youth were empowered to not only identify health threats that are relevant to their lives but also to examine the relationship of these with their own experiences in order to enact change.' p722 [35]

Enhanced ability to connect and communicate with people was viewed as being transferrable to situations beyond the research:

'Youths' reflections that they were able to communicate better with peers and adults and make decisions as a group could be helpful not only in coping with challenging living situations in the shelter but also in successfully navigating larger goals like seeking and maintaining employment or housing.' p386 [32].

Different studies created unique, context-specific relationship impacts, for example, Wartenweiler's [40] research topic, corporal punishment, is linked to relationships within the family, where most participants:

'reported changes in parental discipline and their parent-child relationship after the parents' meeting. . ... Arlyn has since returned home to live with her mother, and their

communication has improved. She reported: 'I am not scared anymore because I know my mother is now listening to my problems and to my feelings' (Arlyn, session 15). p418 [40].

Within this study the citizen scientists were both the researchers and the subject of the research, which may enhance the impacts on relationships given the meaningful nature of the topics.

Participants indicated that they were interested continuing the research themselves or pursuing avenues linked to the research [42,46].

'Others decided to pursue their work with local authorities after their participation in the project had ended to implement their policy ideas.' [46].

## Discussion

This paper presents, to our knowledge, the first evidence synthesis that systematically reviews and thematically analyses qualitative research regarding young citizen scientists' participant experience. 33 papers were included in the review. From this synthesis, six themes describe the participant experience: Personal growth, relationships, power, self-efficacy, communication, and decision-making. These themes offer a new insight into citizen scientists' experiences, allowing researchers to begin to understand how they can impact/develop this experience, resulting in potential benefits for the participant and the research. Despite the common themes identified, for each young citizen scientist, the experience was unique, and situated within a specific context.

Updating the synthesis yielded the discovery of an additional 10 papers over the span of 16 months. Over the preceding decade, only 23 papers had been identified, which underscores the significance of participant experience as a growing area of interest. The update identified further evidence to refine, support and give depth to our themes and enriched the definitions of the aspects surrounding the themes. One notable addition was COVID-19 as a wider context factor influencing participant experience [36,44,46,50]. Given the specific timing of the COVID-19 pandemic, its impact on research projects had not been extensively documented prior to 2022.

### Participant experience

Previous literature synthesising participant outcomes has not been youth-focused and has been primarily focused on the citizen scientists' learning outcomes [3,12,19,62] such as topic-specific knowledge [12]. This element of the experience was also identified through the current synthesis within the theme 'personal growth', which encapsulates an individual's skills, knowledge, confidence, and emotional development. One area that may be of specific relevance to young people was the idea that citizen scientists began to see the system as weighted against them [53]. Makuch [63] suggests citizen science can empower children to contribute to local and global issues. However, at such a key stage of development, if projects are not defined with clear outcomes, they can cause disillusionment for the young citizen scientists [53,64].

Power and relationships were conceptualised as key dimensions of youth participant experience which have been identified as benefits for the citizen scientists within citizen social science [65]. This finding could be due to the inclusion of YPAR studies in the synthesis, as previous research focused on learning outcomes [3,5,12,62] is specific to studies designed, from conception, as citizen science projects. Agency, leadership, social and interpersonal skills are cited as key outcomes for youth taking part in YPAR studies [66]. Additionally, the inclusion of power and relationships as prominent parts of participant experience may be attributed to the specific focus of this synthesis on young people's experiences [3,12]. Children and young people have different neurocognitive development to adults resulting in different

priorities and personality traits [67] which could also contribute to the broad range of experiences. With this in mind, understanding the experience of young people synthesised in this paper will enable future researchers to understand what experiences are like for citizen scientists, and how they may be improved.

The synthesis suggests the exciting potential for citizen science to develop the young people taking part, not only relating to scientific learning, and improvement of research skills supported by citizen science literature [12], but also in terms of communication, empowerment, and advocacy, supported in the current YPAR literature [66]. The findings on self-efficacy and personal empowerment are important both for the participants and broader society. Personal growth was associated with positive experiences for the young citizen scientists [29,31,32,34,35,40,48,51]. Additionally, improved power relations are linked to societal improvement, with potential benefits for both young people and the community [68,69]. Going forward, citizen science projects need to consider and evaluate the young person's experience and what they are taking away from the project as this could have as much of an impact as the research findings themselves.

Where self-efficacy has been reviewed within citizen science, its focus has been on scientific capability [70,71] and often, it is not reported on at all [62], however, Phillips (19) identified it as an important learning outcome albeit with a narrow focus on science and the environment. This synthesis suggests self-efficacy achieved through citizen science is also related to achieving personal goals and wider societal changes. It has also been reported that citizen scientists who improve in self-efficacy are more likely to continue their citizen science volunteer work in the future [72]. Going forward to recruit, retain and provide young people with a positive experience, the young people's self-efficacy in relation to the wider world as well as within the research could be considered and used as a measure to evaluate the success of a project.

Relationships in the general citizen science literature often refer to the relationship with science rather than the relationship between peers, researchers and others involved in the project, which are key to the child's experience [63]. Relational empowerment is key to children increasing their networks or 'bridging worlds', which can bridge social divisions and improve children's ability to collaborate [70]. Planning space and time for these relationships to develop should be integral to citizen science projects involving young people to improve their experience.

Within this synthesis, communication is seen as a connector between personal growth and relationships within the project. Within the wider literature, communication and citizen science are more closely associated with how the research is communicated to an outside audience, including storytelling and using the media [14]. Communication is a vital part of citizen science both within the citizen science project and in explaining the research to an outside audience, potentially improving the link between science and society [73]. Skarlatidou [74] evidences the stakeholders involved in citizen science. For example, civil society organisations, government agencies and academic organisations. Understanding the communication between these stakeholders and young participants is an interesting area of future research.

Decision-making within citizen science is complex and covers many different scenarios within citizen science [75]. When working with young citizen scientists, this complexity increases, and the power difference between children and adults is an ethical challenge for researchers [76]. One way this challenge can be seen is reflected in who can make decisions. One of the notable ways children can exercise their power is to leave a citizen science project [32,41]. However, with citizen science being used more frequently within compulsory education settings, there is a danger that children will be 'volunteered' for projects by school staff. This could result in their agency being diminished [77], this needs to become an item of discussion within the citizen science community.

The participant experience was broken down and discussed as separate themes throughout the findings. However, Wagaman's [61] paper describes part of the participant experience, which was important to the citizen scientist, feeling connected to a researcher who she thought was transgender like herself. When the citizen scientist and the researcher met, the citizen scientist discovered she was a cisgender female. This specific experience of feeling connected to the facilitator because of her profile and then having to readjust when they met, in reality, does not fit easily into a specific theme. Yet, it is still a part of this young citizen scientist experience. This example emphasises how context-specific these experiences can be and suggests that the elements of participant experience are unique to each young participant in each research study.

## Factors influencing experience

Exploring factors influencing participant experience was not the primary aim of this synthesis. However, through inductive coding this synthesis has identified additional important key elements that need to be considered to create an environment that would foster a positive participant experience.

Project design plays an important role. A certain condition (such as project length) does not always result in the same outcome, as contextual factors such as the people involved, the setting, and the research topic also play a significant role in shaping whether and how factors influence participant experience [78]. Thus, these factors are not a 'checklist' to design a positive participant experience, but rather, it will be up to future researchers and citizen scientists to create projects that nurture a positive experience.

Considering the factors influencing the experience is a key aspect of both understanding and helping to foster a positive experience within a citizen science project. The factors influencing experience are similar to the dimensions of critical youth empowerment [69]. The study lists equitable power-sharing between youth and adults, which resonates with the factors influencing power. Jennings's [69] work has been used as a guide to foster and evaluate empowerment in interventions [64,79,80]. Findings of this synthesis could guide citizen science projects to improve and evaluate participant experience.

## Wider impacts

This study identified key wider impacts such as personal development, improved relationships, self-efficacy, self-advocacy, and critical consciousness. Personal development included elements such as educational, social, and personal growth of the young citizen scientists that could remain after the project. The wider literature discusses increasing scientific literacy [63,75] alongside the potential for self-efficacy and self-advocacy through participation in citizen science [71,79]. However, more evidence is needed to support sustained change beyond the project in young people, highlighting an area for future research.

Within the findings, critical consciousness was said to give youth an understanding of injustice [35,38,53]. However, as discussed within the power theme, if the citizen scientists' research was not acted on, the participants were disheartened by the lack of change [53]. Khan's (2022) paper highlights the need to move beyond critical consciousness to enable child-led knowledge, collaboration, and support to challenge 'the white institutional worlds' by which young people are governed rather than simply understanding and raising awareness [79]. Future projects must understand and plan for a positive participant experience, specifically when including multicultural youth, to ensure that young people are not burdened by cultural labour.

## Limitations

A key limitation of the synthesis is that although all studies included met the eligibility of being 'citizen science' projects, several methodological approaches were included, such as YPAR and community research. The drawback to this is that the research team has classified the study as citizen science rather than seeking the opinion of the authors of the studies—the study hadn't been designed from the start with characteristics and principles of citizen science in mind [10,22]. Although participant experience was discussed in each of the papers selected, the aim of the study was not always to evaluate participant experience, which is a key citizen science principle [10]. The methods used within the synthesised studies and the quality of analysis also varied.

The search terms for this synthesis included children and young people (Mean age under 25). The 33 included papers consisted of studies with citizen scientists aged 9 and upwards, mainly teenagers and young adults. The experience of younger children taking part in a research project will be very different from that of someone in their early 20s, and this needs to be explored.

Despite the common themes identified, for each young citizen scientist, the experience was unique and situated within a specific context. While beneficial for synthesising complex information, thematic analysis faces limitations in capturing themes' interconnected and independent nature, making it difficult to convey these relationships effectively. Although this is represented in Fig 2. it is difficult to express this through written narrative. The impact of any single factor will be related to and dependent on other factors, motivations, context, and the individual.

The research followed a robust methodology a systematic approach to identifying papers, followed by independent screening and critical appraisal by a team of reviewers. Nevertheless, it is important to acknowledge that qualitative synthesis invites subjectivity and is interpretive in nature. To ensure the credibility of our synthesised themes, we continuously referred back to the original articles and took a collaborative approach to reach conclusions.

## Strengths of the project and future suggestions

This synthesis offers researchers a comprehensive understanding of the experiences of young citizen scientists. Examining various studies collectively provides a more inclusive perspective on these experiences. For instance, it combines the experiences of indigenous groups, youth of colour, transgender individuals, and studies involving both single and mixed genders. This approach aims to uncover the common experiences shared by young people engaged in citizen science activities.

A key principle of citizen science is that citizen scientists and researchers benefit from participating [10]. Future research could compare the participant experience within specifically designed citizen science projects with the experience of participants from different participatory approaches [81,82]. Additionally, the studies synthesised fall under different typologies of citizen science studies [83,84] and different disciplines. Creating a synthesis that considers the various types of citizen science and if they are from the social or natural sciences could allow for further comparisons and new insights into young people's experiences.

Numerous citizen science studies were not included in this synthesis as participant experience was not reviewed or evaluated within them. Some studies within the synthesis do not have participant experience as a primary aim. This urgently needs to be addressed to ensure young people benefit from giving their time to create new knowledge through research.

Kieslinger's [81] framework for evaluating citizen science projects emphasises scientific impact, learning and empowerment of participants and the wider impact on society.

Participant experience is intrinsically linked to these key elements [81] and should be used to judge the success of a citizen science project. Future citizen science projects, should include post project follow ups to assess long term impacts, enhancing our understanding of how citizen science affects children and young people.

## Key conclusions

This research is the first qualitative evidence synthesis, to our knowledge, that has reviewed and thematically analysed qualitative research discussing the participant experience of young citizen scientists. 33 papers were reviewed. The key takeaways are the themes of power, relationships and personal growth interlinked by communication, self-efficacy, and decision-making. It stresses the importance of including children and young people in research that impacts them, without causing harm. By focusing on the distinct needs and impacts of young people in citizen science, this study sets a foundation for future research to enable projects beneficial for young participants.

## Supporting information

**S1 Checklist. Prisma check list.**
(DOCX)

**S1 Table. ENTREQ guidance report.**
(DOCX)

**S2 Table. Key search terms.**
(XLSX)

**S3 Table. Summary of studies.**
(XLSX)

**S4 Table. Quality assessment and methods of investigating participant experience.**
(XLSX)

**S5 Table. Codebook.**
(XLSX)

## Acknowledgments

The authors are grateful to the young citizen scientists who participated in the original primary research, which has been synthesised.

We would like to thank Jennifer Rowland the librarian who supported us with our search.

The update to the synthesis could not have been conducted without the assistance of Kiran Fatima and Ahlam Sawsaa.

## Author Contributions

**Conceptualization:** Marie T. Frazer, Michael J. Tatterton, Jen Hall.

**Data curation:** Marie T. Frazer, Amy Creaser, Jen Hall.

**Formal analysis:** Marie T. Frazer, Amy Creaser, Jen Hall.

**Funding acquisition:** Jen Hall.

**Methodology:** Marie T. Frazer, Amy Creaser, Jen Hall.

**Supervision:** Michael J. Tatterton, Andy Daly-Smith, Jen Hall.

**Writing – original draft:** Marie T. Frazer.

**Writing – review & editing:** Marie T. Frazer, Amy Creaser, Michael J. Tatterton, Andy Daly-Smith, Jen Hall.

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
