## [Decision Letter · Decision Letter 0]

10 Jan 2024

PONE-D-23-26515Exploring children and young people’s experience of participating in citizen science – a qualitative evidence synthesis.PLOS ONE

Dear Dr. Frazer,

Thank you for submitting your manuscript to PLOS ONE. After careful consideration, we feel that it has merit but does not fully meet PLOS ONE’s publication criteria as it currently stands. Therefore, we invite you to submit a revised version of the manuscript that addresses the points raised during the review process.

We look forward to receiving your revised manuscript.

Kind regards,

Allanise Cloete, PhD (Anthropology)

Academic Editor

PLOS ONE

Journal Requirements:

2. Please be informed that funding information should not appear in the Acknowledgments section or other areas of your manuscript. We will only publish funding information present in the Funding Statement section of the online submission form. Please remove any funding-related text from the manuscript.

3. Thank you for stating the following financial disclosure: "The author(s) received no specific funding for this work."

**Additional Editor Comments:**

• This is an important topic and has significant implications for research that is impactful, in the lives of young people. However, there are a few unnecessary errors which warrants a resubmission.

• Why has the librarian not been included as an author or in the least official acknowledgement by name in the acknowledgement section?

o “The research team (MF, JH, ADS, MT) and librarian developed 147 search strategies (S2 Key search terms)”

• Please mention the 7 databases in the body of the manuscript – and why you have selected these 7 databases – either providing a reason for each – or an overall reason

• Which 7 databases? And why these 7 databases?

• Different fonts used in the manuscript – looks like Times New Roman and Arial –

• Duplication of paragraphs – just before the conclusion

Reviewers' comments:

Reviewer's Responses to Questions

**Comments to the Author**

1. Is the manuscript technically sound, and do the data support the conclusions?

Reviewer #1: Yes

Reviewer #2: Partly

2. Has the statistical analysis been performed appropriately and rigorously? 

Reviewer #1: N/A

Reviewer #2: N/A

3. Have the authors made all data underlying the findings in their manuscript fully available?

Reviewer #1: Yes

Reviewer #2: No

4. Is the manuscript presented in an intelligible fashion and written in standard English?

Reviewer #1: Yes

Reviewer #2: Yes

5. Review Comments to the Author

Reviewer #1: The manuscript under review explores a topical subject: the experience of young people participating in citizen science. This inquiry is crucial in light of the escalating adoption of citizen science projects in contemporary research. The methodology, a thematic synthesis of research on citizen scientists' experience fills a pending research gap. It emphasizes the need for Citizen Science projects to monitor participants experience as a crucial factor for overall success and long-term impact of CS.

The methodology section elaborates on the qualitative thematic synthesis used for collating literature on the young citizen scientists' experience. The thorough search across seven databases, along with an update in May 2023, indicates a robust and systematic literature search.

However, there are areas of concern. Assigning work steps to authors in the main text is unusual. The inclusion of CRediT (Contributor Roles Taxonomy) or a footnote indicating author contributions might be more suitable and align better with common practice.

It's apparent that the researchers successfully identified several themes representing the participant experience. The paper convincingly highlights the interconnectedness of the identified themes.

However,

- The presentation of factors and their frequency across the identified studies would offer clarity. A tally of how many studies identified factors like "Power and Relationships" would be insightful.

- The categorization of the relevant factors by the type of Citizen Science approach (YPAR, PAR, CBR etc.) could enhance the understanding of the results. This would offer additional interpretations, e.g. as to where power is playing a more important role for overall experience.

- Further information regarding the nature of the CS-projects (natural/technical vs. humanities/social sciences) would be beneficial, especially in the context of the Citizen Social Science (CSS) discourse, which this paper touches implicitely by the selection of studies.

- The illustrations in Skarlatidou et al. (2019), for example, could provide suggestions for displaying results in comprehensive tables.

- The authors might want to check the papers listed below, especially since they directly relate to the domain of CSS, to ensure that the manuscript is well-informed and enriched:

1) Göbel, C., Mauermeister, S., & Henke, J. (2022). Citizen Social Science in Germany—Cooperation beyond invited and uninvited participation. Humanities and Social Sciences Communications, 9(1). https://doi.org/10.1057/s41599-022-01198-1

2) Albert, A., Balázs, B., Butkevičienė, E., Mayer, K., & Perelló, J. (2021). Citizen Social Science: New and Established Approaches to Participation in Social Research. In K. Vohland, A. Land-Zandstra, L. Ceccaroni, R. Lemmens, J. Perelló, M. Ponti, R. Samson, & K. Wagenknecht (Hrsg.), The Science of Citizen Science (S. 119–138). Springer International Publishing. https://doi.org/10.1007/978-3-030-58278-4_7

3) Tauginienė, L., Butkevičienė, E., Vohland, K., Heinisch, B., Daskolia, M., Suškevičs, M., Portela, M., Balázs, B., & Prūse, B. (2020). Citizen science in the social sciences and humanities: The power of interdisciplinarity. Palgrave Communications, 6(1), 1–11. https://doi.org/10.1057/s41599-020-0471-y

4) Skarlatidou, A., Suškevičs, M., Göbel, C., Baiba Prūse, Prūse, B., Tauginienė, L., Mascarenhas, A., Marzia Mazzonetto, Mazzonetto, M., Sheppard, A., Barrett, J., Muki Haklay, Haklay, M., Baruch, A., Moraitopoulou, E.-A., Austen, K., Kat F. Austen, Baïz, I., Berditchevskaia, A., … Wyszomirski, P. (2019). The Value of Stakeholder Mapping to Enhance Co-Creation in Citizen Science Initiatives. 4(1). https://doi.org/10.5334/cstp.226

List of recommendations

Major revisions:

1. Reconsider the presentation of work steps by authors in the main text, possibly using CRediT or a separate declaration.

2. Enhance the results section with a clear presentation of factors and their occurrence in the studied articles.

3. Provide a breakdown of the factors based on the type of Citizen Science approach.

4. Delve deeper into the nature of the CS-projects to ascertain the balance between natural/technical and humanities/social sciences.

Minor revisions:

1. Rectify the presentation error in lines 339-40, ensuring consistency in quotation formatting.

2. Attend to the typographical issues in lines 626-30.

3. Remove repetitions in lines 1002-3.

4. Evaluate potential areas in the text where the content could be made more concise.

5. Consider incorporating insights from the suggested papers on citizen social sciences.

Overall Impressions

The study's foundation is very solid, promising a valuable contribution to the literature on citizen science with young participants. Addressing the aforementioned major and minor concerns will significantly enhance the manuscript's quality and value. I look forward to witnessing this research's advancements, given its potential to shape the future trajectory of citizen science endeavors.

In conclusion, with some revisions, the paper has the potential to be a notable publication in the realm of citizen science research.

Reviewer #2: Thank you for submitting this revised paper with more recent sources. it is true that there are very few synthesis papers on

youth experiences in citizen science, so this is a novel contribution to the literature base. I have several, mostly minor, suggestions to improve the paper before publication.

A transition sentence is needed to introduce the concept of participant experience on line 81. A bit of theoretical framing on what is meant by "participant" and "experience" would be helpful.

You write: "The study method must include a qualitative review of the participants' citizen science experience, either by the citizen scientists themselves or researchers." Can you explain why this was a inclusion criteria? At least some type of justification for why quantitative studies would not be included would be helpful.

Consider some copy editing before publication. For example, many sentences such as this one have awkward grammatical structures: “For the updated search papers were line-by-line coded by two researchers (MF 100%, 210 AS 50%, KF 50%).” Or this: “This study defines relationships as the connectedness and interactions between people and meanings attached to this human connection can be a positive side of participant experience (48,57)”

Could you provide an intercoder reliability for the team coding? See this paper for example: https://journals.sagepub.com/doi/10.1177/1609406919899220

Do you have a reference for this statement or are you hypothesizing that this is how the system works? “As citizen scientists developed skills, knowledge and confidence and emotional understanding, their ability to communicate and interact within relationships changed, which in turn affected the power they felt and were able to use, demonstrated through decision-making.”

On line 478 you wrote “Positive relationships with the facilitators and peer researchers often supported each other and led to positive relationships with the other stakeholders.” What about the negative relationships that were described a few paragraphs earlier – how did those change the nature of the experience?

The communication theme seems very weak and in need of further detail and justification to include in the argument.

The section entitled “wider impacts” seems like a catch all of miscellaneous topics that is difficult to interpret. I suggest either deleting that from the manuscript or trying to recode into some of the other interrelated categories.

I don’t think this statement is accurate: "As with power, relationships have not been highlighted or synthesised as an element 872 of the citizen science experience." Phillips et al (2019) describes this in adults and I believe there is a paper on Monarch Watch which describes this in youth.

A whole paragraph is repeated (lines 1016 – 1021).

Figure 1 was so blurry, I could not read it. In fact, all of the figures were quite blurry. In terms of the venn diagram in Figure 2, could you provide some analytical evidence for these interrelationships beyond what is in the narrative. For example, could you provide a table showing the associations between how often communication came up between relationships and personal growth. I know it's subjective, but by putting a figure that delineates these interrelated relationships, I would want to know how strong that relationship is. Was it mentioned by one study or multiple?

I appreciate all of the supporting documents but what I didn’t see was a codebook – I think this would be the most critical supplementary information to be able to replicate such a study. Could you please provide the full codebook you used to define and categorize your themes?

6. PLOS authors have the option to publish the peer review history of their article (what does this mean?). If published, this will include your full peer review and any attached files.

Reviewer #1: No

Reviewer #2: No

---

## [Author Response · Author response to Decision Letter 0]

2 Feb 2024

Reviewer 1, Thank you for your kind words and identifying the need for the paper in your review, we are passionate about young people in citizen science and to have the need for this paper recognised was uplifting. We are proud that our thorough search and update were noted. We greatly appreciate the time you have taken to suggest improvements. We enjoyed reading the suggested papers, particularly those related to citizen social science and stakeholders involved in citizen science. These recommendations have not only helped improve this paper but we will continue to use the work discussed and ideas proposed in future work, we are grateful you took the time to include these papers. The revisions you have suggested have been addressed below. 

Reviewer 2, Thank you for acknowledging the work that updated the synthesis, and supporting the need for the paper. We appreciate the time you have taken to diligently read through the paper and make your useful comments. Your suggestions have helped improve the flow of the paper. We have also specifically revisited the communication theme to strengthen this section, and the wider impacts theme to clarify what this section is for. The inclusion of a code book we hope will add to the transparency of the work.

---

## [Decision Letter · Decision Letter 1]

5 Mar 2024

PONE-D-23-26515R1Exploring children and young people’s experience of participating in citizen science – a qualitative evidence synthesis.PLOS ONE

Dear Dr. Frazer,

Thank you for submitting your manuscript to PLOS ONE. After careful consideration, we feel that it has merit but does not fully meet PLOS ONE’s publication criteria as it currently stands. Therefore, we invite you to submit a revised version of the manuscript that addresses the points raised during the review process.

We look forward to receiving your revised manuscript.

Kind regards,

Allanise Cloete, PhD (Anthropology)

Academic Editor

PLOS ONE

Reviewers' comments:

Reviewer's Responses to Questions

**Comments to the Author**

1. If the authors have adequately addressed your comments raised in a previous round of review and you feel that this manuscript is now acceptable for publication, you may indicate that here to bypass the “Comments to the Author” section, enter your conflict of interest statement in the “Confidential to Editor” section, and submit your "Accept" recommendation.

Reviewer #1: All comments have been addressed

Reviewer #2: All comments have been addressed

2. Is the manuscript technically sound, and do the data support the conclusions?

Reviewer #1: Yes

Reviewer #2: Yes

3. Has the statistical analysis been performed appropriately and rigorously? 

Reviewer #1: N/A

Reviewer #2: N/A

4. Have the authors made all data underlying the findings in their manuscript fully available?

Reviewer #1: Yes

Reviewer #2: Yes

5. Is the manuscript presented in an intelligible fashion and written in standard English?

Reviewer #1: Yes

Reviewer #2: Yes

6. Review Comments to the Author

Reviewer #1: Dear Authors,

The integration of the reviewers' comments into the revised manuscript is commendable. The approach and methodology employed in the study are robust, aligning well with standard practices within the field. However, there are areas, in particular the discussion, in the manuscript that warrant further refinement to fully realize the impact of this research.

1. Results Section:

- To improve the manuscript's readability and cohesiveness, I suggest integrating the subsection titled "Factors Influencing Power" directly into the main "Power" section. Applying this restructuring across other sections as well will ensure that the discussion of each dimension and its influencing factors is more interconnected and accessible to the reader.

- Figure 2: While the themes in the figure are clear, the selection of factors influencing experience and the broader impacts outlined requires further clarification. As this figure is in the beginning of the results section, its basis of selection remains unclear. Addressing selection of most relevant aspects and potential omissions, especially in the "factors influencing experience" category, and elucidating on the somewhat ambiguous term "wider context" would greatly enhance the reader's understanding. The discussion section would be an appropriate place for adressing this.

2. Discussion Section:

- The current discussion of the results, while comprehensive, could benefit from a more focused and sharpened narrative. I recommend a more detailed exploration of the aspects that are particularly relevant to young participants in citizen science. Specifically, it could be beneficial to:

- Contrast the findings with existing studies that discuss influencing factors irrespective of participant age, and/or

- Highlight elements uniquely pertinent to younger individuals, such as the impact of age restrictions or research topics that resonate more strongly with this demographic.

- Wider Impacts Subsection: The purpose of this subsection seems to be primarily to advocate for standardized evaluations in citizen science projects. While important, this could be succinctly addressed in the conclusions. Instead, a justification for the chosen aspects of wider participation as depicted in Figure 2 would be more beneficial in this context.

- Concentration on Limitations**: The "Strengths and Limitations" section should be streamlined to emphasize the limitations of the methodological approach. This focus will provide readers with a clearer understanding of the study's scope and the reliability of its findings.

- Conclusive Summary: A concluding section that precedes or is integrated with "Future Suggestions" appears necessary to succinctly summarize the key messages derived from the analysis of youth experiences in citizen science. Highlighting some of the strengths mentioned previously, along with a compelling articulation of how young people represent a distinct group with specific needs and impacts within citizen science, will lay a solid groundwork for future research.

3. Minor Issues:

- Table 1 Presentation: The right side of Table 1 is truncated, which may hinder the comprehension of the data presented. A review and adjustment of the table's layout are recommended.

- Legibility of Figures: Figures 1 and 2 suffer from legibility issues, potentially due to the PDF export process. Ensuring the clarity and readability of these figures is crucial for readers to fully grasp the presented data and analyses.

By addressing these issues, I am confident that your article will significantly influence further research on this topic.

Reviewer #2: Thank you for addressing each of the reviewer comments. I am satisfied with the diligence the authors took to improve the paper.

7. PLOS authors have the option to publish the peer review history of their article (what does this mean?). If published, this will include your full peer review and any attached files.

Reviewer #1: **Yes: **Justus Henke

Reviewer #2: No

---

## [Author Response · Author response to Decision Letter 1]

25 Mar 2024

Please find attached the 'Response to Reviewer file as requested. Thank you for taking the time to consider our paper.

---

## [Editor Report · Decision Letter 2]

6 May 2024

PONE-D-23-26515R2Exploring children and young people’s experience of participating in citizen science – a qualitative evidence synthesis.PLOS ONE

Dear Dr. Frazer,

Thank you for submitting your manuscript to PLOS ONE. After careful consideration, we feel that it has merit but does not fully meet PLOS ONE’s publication criteria as it currently stands. Therefore, we invite you to submit a revised version of the manuscript that addresses the points raised during the review process.

**ACADEMIC EDITOR:**It is commendable how well you had responded to the reviewer comments. However, it is recommended to proofread before resubmitting.Please also refer to PLoSONE's reference style guide:In-text citationsReferences should be cited in the text by sequential numbers in square brackets:This sentence cites one reference [1].This sentence cites two references [1,2].This sentence cites four references [1–4].Please ensure that your decision is justified on PLOS ONE’s publication criteria and not, for example, on novelty or perceived impact.

We look forward to receiving your revised manuscript.

Kind regards,

Allanise Cloete, PhD (Anthropology)

Academic Editor

PLOS ONE
---

## [Author Response · Author response to Decision Letter 2]

10 May 2024

Dear Dr Cloete 

Thank you, Dr Cloete, for your quick response and for providing feedback on our manuscript. 

We are pleased that our responses to the reviewers' comments were deemed commendable. We understand the importance of ensuring clarity and accuracy in our manuscript, and it has been thoroughly proofread, with changes visible in the track changed manuscript. We appreciate the reminder to adhere to PLOS ONES referencing style and have amended the in-text citations. 

We are committed to meeting the journal requirements. As such, we have double-checked the reference list to ensure it is complete and correct and made some modifications to ensure it conforms to PLOS ONE's requirements. Sight corrections were made to references 2, 6, 7, 10, 38, and 56. This is evidenced in the tracked changes document. We apologise that these references may not have met PLOS ONE's requirements. 

Again, Thank you for your valuable feedback and for considering our manuscript for publication in PLOS ONE. 

Sincerely,

Marie Frazer

---

## [Editor Report · Decision Letter 3]

22 May 2024

Exploring children and young people’s experience of participating in citizen science – a qualitative evidence synthesis.

PONE-D-23-26515R3

Dear Dr. Frazer,

We’re pleased to inform you that your manuscript has been judged scientifically suitable for publication and will be formally accepted for publication once it meets all outstanding technical requirements.

Kind regards,

Allanise Cloete, PhD (Anthropology)

Academic Editor

PLOS ONE
---

## [Editor Report · Acceptance letter]

3 Jul 2024

PONE-D-23-26515R3 

PLOS ONE

Dear Dr. Frazer, 

I'm pleased to inform you that your manuscript has been deemed suitable for publication in PLOS ONE. Congratulations! Your manuscript is now being handed over to our production team.

Kind regards, 

on behalf of

Dr. Allanise Cloete 

Academic Editor

PLOS ONE